# Antioxidant Capacity and Phenolic and Sugar Profiles of Date Fruits Extracts from Six Different Algerian Cultivars as Influenced by Ripening Stages and Extraction Systems

**DOI:** 10.3390/foods10030503

**Published:** 2021-02-26

**Authors:** Malika Tassoult, Djamel Edine Kati, María África Fernández-Prior, Alejandra Bermúdez-Oria, Juan Fernandez-Bolanos, Guillermo Rodríguez-Gutiérrez

**Affiliations:** 1Laboratoire de Biochimie Appliquée, Département des Sciences Alimentaires, Faculté des Sciences de la Nature et de la Vie, Université de Bejaia, Bejaia 06000, Algeria; tassoult_malika@yahoo.fr (M.T.); djamelkati@yahoo.fr (D.E.K.); 2Instituto de la Grasa, Consejo Superior de Investigaciones Científicas (CSIC), Campus Universitario Pablo de Olavide, Edificio 46, Ctra. de Utrera, km 1, 41013 Seville, Spain; mafprior@ig.csic.es (M.Á.F.-P.); aleberori@ig.csic.es (A.B.-O.); j.fb.g@csic.es (J.F.-B.)

**Keywords:** *Phoenix dactylifera* L., secondary dates, phenolic profile, antioxidant activities, sugars

## Abstract

The study investigated the phenols, sugar and the antioxidant capacities of date fruit extracts obtained by organic solvents and by hydrothermal treatment from six different Algerian cultivars at two ripening stages for the first time. The analyzed cultivars exhibited potent antioxidant properties (ferric reducing antioxidant power (FRAP), 1,1-Diphenyl-2-picrylhydrazyl (DPPH) and 2,2′-azino-bis(3-ethylbenzothiazoline-6-sulfonic acid (ABTS) scavenging capacities) and different phenols regardless of the solvents and the maturity stages. About 18 phenols were identified and quantified, mainly in the hydrothermal extracts. The earlier stages were characterized by high amounts of o-coumaric acid, cinnamic acid and luteolin, with a noticeable absence of quercetin. The tamr stage presented the highest sugar content (78.15–86.85 mg/100 mg dry weight (DW)) with an abundance of glucose. Galactose was present only in some cultivars from the kimri stage (tamjouhert). Uronic acids were mostly detected at the tamr stage (4.02–8.82 mg gallic acid equivalent/100 mg dried weight). The obtained results highlight the potential of using date fruit extracts as natural antioxidants, especially at industrial scales that tend use hydrothermal extraction.

## 1. Introduction

Date palm (*Phoenix dactylifera* L.) is a major fruit tree in most of the Arabian Peninsula and it is considered one of the most important commercial crops [1,2]. Date fruit is a highly nutritious food product that is rich in simple sugars such as glucose and fructose (65–80%). It constitutes a good source of fibers, essential minerals and vitamins. It is characterized by low amounts of fat and protein with no starch [3,4]. Besides nutritional value, date fruits are rich in bioactive compounds possessing various biological properties [5]. Phytochemical investigations have revealed that these fruits contain secondary metabolites such as phenolic acids, flavonoids, tannins, terpenoids, alkaloids, saponins, sterols or glycosides that are known to have in vitro antioxidant, antimicrobial, anti-inflammatory, anti-proliferative and enzymatic activities [5,6,7].

As they ripen, date fruits go through four ripening stages termed kimri, khalal, rutab and tamr [8]. These represent the immature astringent green, crunchy yellow, soft brown and hard raisin-like stages of development, respectively [9]. During date ripening, a series of metabolic and physiological changes occur: the sugar content increases from the kimri to the tamr stage [10,11], while the phenolic fraction, mainly tannins, decline progressively [12].

Indeed, the increasing assessment on date fruits is a matter of interest due to their bioactivities that rely mainly on the extracted bioactive compounds [4,13]. This is influenced by the ripening stage and the extraction conditions, including the extraction solvent, the extraction method and the ratio of sample:solvent [2,14]. In this context, the highest levels of total phenolic and condensed tannins contents were found in methanol:water 80:20 *v*/*v* date fruit extracts, while the highest total flavonoid contents and antioxidant activities were found in acetone:water 70:30 *v*/*v* [4]. Similarly, extracts obtained from hydrothermally treated date fruits had higher phenolic contents and stronger antioxidant activities [7]. Even though a growing body of literature has highlighted the impact of extraction solvent or the ripening stage on the antioxidant capacities of date fruits, none of them investigated the effect of maturity status and extraction solvent simultaneously on their phenolic pattern, sugar profile and antioxidant capacities. Hence, the current study aimed at screening the proximate composition of some Algerian secondary date varieties from the two last ripening stages (rutab and tamr) which are edible, at evaluating their in vitro antioxidant properties and at investigating the correlations between the antioxidant activity and the total phenols. To our knowledge, this study reports, for the first time, the compositional changes and the evolution of antioxidant activity during ripening. This may help us to understand the biochemistry of the fruit ripening and the major pathways.

## 2. Material and Methods

### 2.1. Chemicals

Acetone (99.78%), methanol (99.90%), sodium carbonate (NaCO3 99.5%), sodium nitrate (NaNO_3_ 99.0%), aluminum chloride (AlCl3 97.0%), sodium hydroxide (NaOH 97.0%), potassium ferricyanide (K_3_(Fe(CN)_6_) 99.0%), iron trichloride (FeCl_3_ 98.0%), iron II chloride (FeCl_2_ 98.0%) and trichloroacetic acid (TCA 98.0%) were from Biochem Chemopharma (Georgia, USA). Folin–Ciocalteu reagent was from Biochem Chemopharma (Montreal, Quebec) and 1,1-Diphenyl-2-picrylhydrazyl (DPPH 95%) was from Sigma-Aldrich (Sternheim, Germany). Gallic (99.5%), p-coumaric (98%), ferulic (99%) and caffeic acids (98%), rutin (94%), luteolin (98%), isoquercetrin (97%), quercetrin (95%) and quercetin (98%) were from Sigma-Aldrich Co (Saint-Louis, MO, USA).

### 2.2. Sampling

Six date fruit cultivars (dalt, deglet nour, ghars, tamezwert n’telet, tamjouhert and tazarzeit) at two different ripening stages (tamr and rutab) were harvested from M’zab valley, Ghardaia, Algeria. The chosen fruits had a uniform size and were free of physical damage, insect injury and fungal infection. Upon arrival at the laboratory, the seeds were manually separated from the pulp and hand-cut into small pieces. The samples were maintained at −20 °C before their analyses.

### 2.3. Extraction Procedure of Bioactive Compounds 

The bioactive compounds were extracted using three solvents: methanol:water (65:35 *v*/*v*), acetone:water (65:35 *v*/*v*) and water. The extraction was carried out according to the method of Al-Zoreky and Al-Tahar [15] with slight modifications. Briefly, five grams of each date cultivar from each stage were homogenized with 20 mL of each solvent using an Ultra-Turax T25 homogenizer (IKA-labortechnik, Germany) for 2 min. Only the aqueous extracts were sterilized for 1 h at 120 °C to obtain hydrothermal extracts. Subsequently, the MeOH:H_2_O extracts (ME), acetone:H_2_O extracts (AE) and the hydrothermal extracts (HE) were centrifuged at 4000 g (Sigma 2-16K, Osterode, Germany) for 5 min, paper-filtered and, finally, stored at 4 °C and processed for analysis within 1 week. 

### 2.4. Determination of Uronic Acid Content

Uronic acid content (UAC) was determined by tetraborate-sulphuric acid colorimetric assay using the m-hydroxyphenyl reactive at 520 nm in an iMarkTM microplate absorbance reader [16]. The results were expressed as milligrams of galacturonic acid equivalents per 100 mg of dry weight (mg GAE/100 mg DW).

### 2.5. Determination of Sugar Content

According to Mrabet et al. [17], the anthrone-sulfuric acid colorimetric method was used to determine the total sugar content (TSC) at 630 nm in an iMarkTM microplate absorbance reader (Bio-Rad, Hercules, CA, USA). The results were expressed as milligrams per 100 mg of dry weight (mg/100 mg DW).

### 2.6. Determination of Individual Sugars

Individual sugars were assessed according to Mrabet et al. [17]. Prior to determining the neutral sugars, the sugar fraction was hydrolyzed with trifluoro acetic acid (TFA) for 1 h at 121 °C. The released sugars were reduced and then acetylated and, finally, identified and quantified as alditol acetates using gas chromatography (HP 6890 Plus, Hewlett-Packard, Palo Alto, CA, USA) equipped with a capillary column (30 m × 250 μm × 0.20 mm, SP-2330, Supelco, Bellefonte, PA, USA). The flow of the vector gas (Helium) was 2.2 mL/min at a pressure of 148.24 kPa. The run time was 40.7 min, during which the injector and flame ionization detector (FID) temperatures were 250 and 300 °C, respectively. After 2 min of the injection (splitless mode), the oven temperature was 50 °C; it was then held progressively at 180 °C and finally increased to 220 °C for 22 min. The internal standard was Myo-inositol. The results were expressed as milligrams per kg of dry weight (mg/kg DW).

### 2.7. Determination of Total Phenolic Content 

The total phenolic content (TPC) was determined by the Folin–Ciocalteu spectrophotometric method [18] and was expressed as grams of gallic acid equivalents per 100 g (mg GAE/100 g dry weight (DW)).

### 2.8. Determination of Phenolic Profile Using HPLC-DAD

To determinate the phenols, 20 μL of each extract was injected into HPLC–DAD (liquid chromatography with a diode-array detector). for 55 min. The separation was carried out on a reversed phase equipped with a C-18 column (Teknokroma Tracer Extrasil ODS-2, 250 mm, –4.6 mm, inner diameter i.d. 5 mm). The flow rate of the mobile phase (eluent A = water/trichloroacetic acid 0.01%; eluent B = acetonitrile) was 1 mL/min and its gradient system was 95% A initially, 75% A in 30 min, 50% A in 45 min, 0% A in 47 min, 75% A in 50 min and 95% A in 52 min until completing the run time. Quantification of the identified phenols was performed using external standards’ calibration curves [19]. The results were expressed as milligrams per 100 g of dry weight (mg/100 g of DW). 

### 2.9. Determination of Antioxidant Activity

Prior to determining the antioxidant activity (AOA), the extracts were evaporated under nitrogen. The extracts were then dissolved in water and, finally, paper-filtered (0.45 µm). The resultant extracts were used to assay this activity using different tests.

#### 2.9.1. DPPH Radical Scavenging Capacity

The free radical DPPH-scavenging capacity was assessed as described by Fernández-Bolaños et al. [20]. Briefly, 195 µL of DPPH solution was added to 5 µL of extract. The mixture was incubated in the dark for 30 min at room temperature. The absorbance was recorded, against a blank, at 490 nm (in an iMarkTM microplate absorbance reader). The Efficiency 50 (EC 50) was calculated and was expressed as mg/mL.

#### 2.9.2. ABTS Radical Scavenging Capacity

The 2,2′-azino-bis(3-ethylbenzothiazoline-6-sulfonic acid( ABTS) radical-scavenging capacity’s measurement was also performed to evaluate the antioxidant capacity. This assay was conducted according to the method used by Rubio-Senent et al. [21], with slight modifications. Briefly, 187 µL of ABTS solution was added to 13 µL of extract. After 6 min of incubation, the absorbance was read at 750 nm (in an iMarkTM microplate absorbance reader) against a blank. The results were expressed in terms of the Trolox equivalent antioxidant capacity (TEAC) as mmol Trolox/g of dry extract.

#### 2.9.3. Ferric Reducing Antioxidant Power (FRAP)

The ferric reducing antioxidant power (FRAP) of the date extracts was assessed using the method of Mrabet et al. [19]. Extracts (10 μL) were added to 10 µL of FeCl_3_ (6 mM). The mixture was incubated at 50 °C for 20 min, and then 80 µL of dipyridyl (0.5%) was added. The absorbance was recorded at 490 nm (in an iMarkTM microplate absorbance reader) after 30 min of incubation at room temperature. The results were expressed as mmol Trolox/g of dry extract.

### 2.10. Statistical Analysis

All analyses were carried out in triplicate and the results were expressed as means ± standard deviation. Statistical analyses were performed using Sigmaplot software and the differences at *p* ≤ 0.05 were considered statistically significant. An ANOVA test was performed to compare between the different cultivars, stages and solvents, after which the T-Tukey test was realized to classify the samples into groups. The principal component analysis (PCA) served to visualize the correlation between the bioactive compounds and the antioxidant capacity of our samples.

## 3. Results and Discussions 

### 3.1. Total Sugar Content

The total sugar content was determined colorimetrically and the results are shown in Table 1. The analyzed fruits were characterized by a high TSC that varied significantly among them, from 26.12 ± 1.20 mg/100 (kimri tazarzeit) to 85.89 ± 5.54 mg/100 mg DW (tamr tazarzeit). Moreover, methanolic and acetonic extracts’ TSCs were higher than those of the hydrothermal ones, for which the TSC diminished considerably down to 50% in tamjouhert and tazarzeit cultivars. These results are in agreement with those published previously, in which similar values were reported for other date varieties, whose sugar content varied from 81% to 88% [22] and from 90% to 92% [23].

Noticeably, the TSC of the analyzed cultivars showed an increase from the rutab to the tamr stage. These findings are also supported by the study of Haider et al. [8] who also found that sugar amounts increased progressively as the date fruits matured. Actually, the high sugar content at the latest stage rendered the fruits extremely resistant to microbial spoilage after harvest, which give them a good storability [14].

The diminution of the sugar content in hydrothermal extracts could be explained by the degradation of sugars under higher temperatures. In fact, the degradation process starts at temperatures above 60 °C. This was confirmed in our study by the apparition of some degradation products such as hydroximethylfurfural (HMF) and furfural. In this context, Mrabet et al. [7] reported the diminution of sugar content for three thermally treated date cultivars and the increase in the diminution percentage with the severity of the thermal treatment (140, 160 and 200 °C).

The high amounts of TSC of our cultivars pointed out their nutritional value through their sweetness and their high calories, hence providing the required energy for the organism. 

### 3.2. Uronic Acids Content

The uronic acids content (UAC) (Table 1) showed significant differences between all the analyzed samples; depending on the extraction solvents, the HE had a lower UAC followed by AE and ME. Additionally, for all the extracts used, the earliest stages’ UACs were significantly lower than those of the latest stage. 

Like sugars, the determination of the UAC could be a key component in determining quality. In facts, high pectin content and low lignin content indicate good quality [22]. Mrabet et al. [17] found values ranging between 4.1 and 4.5 g/100 g. On the other hand, Elleuch et al. [24] analyzed two Tunisian cultivars, which were characterized by very low amounts of pectin at around 2%.

The diminution of UAC after the thermal treatment concurs with the results of Mrabet et al. [7] who found that the UAC of three cultivars was affected by the thermal treatment. 

The fruit’s softness during the tamr stage could be used to explain the high UAC during this stage. Actually, the dates’ softening characterizes their ripening process; this is due to enzymatic activities of polygalacturonase that helps to break down the pectin in the cell walls and, therefore, lead to increasing amounts of UAC. Despite the variation in the UAC among cultivars, the results evidenced, to a great extent, the excellent organoleptic quality of the cultivars studied. 

### 3.3. Sugar Profile

There were significant differences in the samples’ individual cell wall sugar compositions (Table 2). The hydrothermal extracts had the lowest amounts in comparison with the other extracts. Additionally, cultivars at the tamr stage had higher amounts than those at the rutab stage. Glucose was the main sugar in all of the samples at all stages, regardless of the extraction procedure. Arabinose and rhamnose were also among the sugars of interest. Fucose was found in lower amounts, while galactose was present only in some cultivars at the kimri stage, such as tamjouhert. Regarding pentoses, mannose and xylose amounts were near 69.61 and 11.75 mg/kg DW in ME, respectively. 

The sugar profile of dry dates has been widely studied. Our results are in agreement with those of Elleuch et al. [24], who reported that date flesh mainly contains glucose (8.8–9.4% DW). However, Mrabet et al. [22] found that xylose was the predominant sugar (50%). Moreover, a similar trend of increasing amounts of reducing sugar (RS) (up to 47%), to the detriment of non reducing sugar (NRS), through ripening is reported by Haider et al. [8]. Likewise, El-Sohaimy [25] and Ziadi et al. [23] highlighted that glucose and fructose have lower quantities at khalal and higher quantities at the tamr stage, whereas sucrose was present only at the khalal and rutab stages. The most likely explanation of the sudden drop in NRS and the increase in RS at the tamr stage is the rising activity of the invertase enzyme that hydrolyzes the sucrose into glucose and fructose [26]. 

Interestingly, the proportion of RS/NRS is commonly used to gain an idea about the softness or dryness of a fruit. The higher the proportion, the softer the fruit [19]. Consequently, the dominance of RS in all the studied samples pointed out that our cultivars belong to the soft date type.

### 3.4. Total Phenolic Content 

Significant differences were easily pointed out between all the analyzed samples (Table 1). It is clearly shown that the TPC of HE is higher than that of AE and ME. Furthermore, the PC was more abundant in the earliest stages.

TPCs found in this work were as high as those reported in the literature, which vary between 95 and 193 mg GAE/100 g DW [5]. Similar values that ranged from 248 to 385 mg GAE/100 g DW were reported by Hachani et al. [27]. Higher TPC values were found by Benmeddour et al. [28], Benkerrou et al. [29] and Djaoudene et al. [30] These authors reported that the TPC reached up to 727.03, 955 and 1393.50 mg GAE/100 g DW, respectively.

The observed variations may mainly be due to the variety, ripeness, cultivation region, environmental factors, storage time and conditions [31]. The amount of sunlight received is critical because it enhances the Maillard reaction, including the reaction between amino acids and RS, where some of the resulting products are phenolic compounds [32].

Other factors including extraction conditions such as solvent and ratio of material/solvent may also be the cause. This was supported by another study in which it was found that the TPC ranged between 20.38 and 69.85 mg GAE/100 g DW in 80% methanolic extracts and between 11.13 and 18.23 mg GAE/100 g DW in 70% acetonic extracts [3]. Increasing TPC in hydrothermal treatments was also reported in the study of Mrabet et al. [19], who mentioned that these amounts increased significantly by 15% with increasing temperature.

Additionally, the TPC decreased significantly from the rutab to the tamr stage. This decrease was also mentioned in previous investigations, which mentioned that TPCs were higher at khalal and thereafter but declined at the fully ripened stage from 50 to 400 mg/100 g, from 2.5 to 0.5 mg/g fresh weight (FW) and from 468.9 to 356.9 mg GAE/100 g DW [9].

### 3.5. Phenolic Profile

The phenolic pattern was studied and the results are mentioned in (Figure 1). The cultivars share mostly the same phenolic profiles with little differences. Tamjouhert and ghars cultivars had the highest levels of p-coumaric acid (0.48 and 0.42 mg/100 g DW, respectively). Nevertheless, the other varieties contained low levels of caffeic acid.

In comparison with ME and AE, the HE had significantly high amounts of individual phenols. Likewise, the samples had different phenolic compound patterns with large variations depending on the cultivars, the solvent and the ripening stage. Briefly, caffeic and cinnamic acids were the most abundant phenolic acids while luteolin and catechin were the most abundant flavonoids.

Regarding the ripening stages, the amounts of luteolin, cinnamic acid and protocatechuic acid decreased considerably in the tamr stage. On the other hand, the quercetin amount increased significantly during this stage.

Beside phenols, some sugar degradation products including furfural (nd) and HMF were found only in TE and their amounts increased noticeably in the tamr stage.

Interestingly, analyzing the polyphenol profiles by HPLC–DAD allowed the quantification of some compounds that have not been previously quantified in Algerian date fruits. Mansouri et al. [33] identified p-coumaric, ferulic, sinapic and cinnamic acid derivatives as well as flavone glycosides, flavanone glycosides and flavonol glycosides in Algerian date fruits from Ghardaia; nevertheless, these phenolics were not quantified. Benmeddour et al. [28] identified and quantified nine free phenolic acids and flavonoids including gallic acid (70–92%), caffeic, p-coumaric and ferulic acids, isoquercetrin (13–51%), rutin (19–40%), quercetrin (16–54%), quercetin (0.64–3.3%) and luteolin (0.6–4.5%). Hachani et al. [27] detected 23 phenolic compounds using LC/MS, most of which are hydroxycinnamic derivatives, formic acid derivatives (caffeic acid-formic acid), ferulic acid-o-hexoside derivatives, asapigenin pentosyl hexoside; phloridzin (phloretin-o-hexoside), luteolin rhamnosyl, dihexoside, isorhamentin hexoside and quercitin-7-glucoside. Furthermore, o-p-coumaroylshikimic was only detected on an infected tinnaser cultivar.

Similar phenolic acid profiles have been reported for Omani cultivars, showing that ferulic acid was the major compound [34]. In their study on the Tunisian secondary date cultivars. Mrabet et al. [16] identified and quantified some phenolic acids, including gallic, protocatechuic, vanillic or p-coumaric acids, and tyrosol, where gallic acid was the main compound.

Regarding the effect of extraction, samples from different solvents had similar phenolic patterns with little differences in their amounts, which were higher in the TE. These results are in disagreement with those of Hachani et al. [27], who found that caffeic acid-o-(sinapoyl-o-hexoside) was detected only in the 80% methanolic extract, whereas luteolin was identified in all 70% acetonic extracts. According to Mrabet et al. [19], gallic acid was the only phenolic compound whose concentration decreased with stream explosion treatment (SET) at the highest temperature. Additionally, Allaith et al. [35] also reported increasing concentrations (from 160.3 to 866.2 mg/kg) of phenols in thermally treated dates at 100 °C. According to these authors, the temperature solubilizes higher quantities of phenols. 

In terms of thermal degradation products, similar results were published by Mrabet et al. [16]. These authors found that HMF ranged from 3687 ± 92 to 4791 ± 50 mg/kg in ST and from 9004 ± 759 to 13,157 ± 664 mg/kg in SET, while furfural dropped down from 1.4 ± 0.1 to 0.1 ± 0.0 mg/kg, probably due to its volatilization under high temperatures

### 3.6. Antioxidant Activity

The analyzed cultivars showed great variation between them in terms of AOA, assayed by three different tests (Figure 2, Figure 3 and Figure 4). The differences are genotype-dependent and influenced by fruit maturation stage and storage time [33]. The soil conditions and doses of fertilizers were also reported to be responsible for affecting the antioxidant efficiency (AOE) of date fruits [8].

The free radical DPPH scavenging ability of the samples was high, with EC 50 values ranging between 5.80 ± 0.08 and 15.91 ± 1.45 mg/mL in HE. Furthermore, the strongest DPPH antiradical efficiency was recorded at the earliest stage in HE, followed by AE and finally ME. 

For the ABTS radical scavenging capacity, the values obtained for HE were significantly higher than those for ME and AE. In comparison with the latest stage, the ABTS results at the earliest stages were significantly the highest.

Regardless of the extraction procedure and the ripening stage, all of the samples exhibited a good reducing power, which varied significantly from 78.70 ± 0.26 to 141.56 ± 0.33 mmol Trolox/kg DW.

Concerning the DPPH antiradical efficiency, all of the extracts, especially HE, exhibited strong scavenging capacity. This finding is in agreement with those of Mansouri et al. [33], who reported that date fruits exhibited potent DPPH scavenging capacities. Similarly, it was reported that DPPH inhibition varied between 32% and 86% [28]. Biglari et al. [31] reported that Iranian dates possessed the highest AOA that ranged from 22.83 to 54.61 μmol trolox equivalent/ 100g of fresh weight (μmol TE/100g FW). In their study about the AOA of Algerian date fruits, Hachani et al. [27] reported that the AOA of some commercial antioxidants, assayed as EC 50 of DPPH, is much stronger than our cultivars (0.004, 0.002, 0.005 and 0.018 g/L for butylated hydroxytoluene (BHT), Vitamin C, tert-Butylhydroquinone (TBHQ ) and Vitamin E, respectively).

Free radicals are not exclusively negatively charged like DPPH, and that is why the determination of the scavenging capacity of ABTS, which is positively charged, is important. According to Mrabet et al. [19], the ABTS antiradical efficiency of the thermally treated dates was similar to BHT—it reached 0.55 mmol Trolox/g. The same authors demonstrated that the AOE increased with the severity of the thermal treatment. 

Regarding reducing power, all cultivars exhibited potent reducing power. These results were in line with those reported for Algerian dates by other authors, which varied from 272 to 1175 mg GAE/100 g DW and from 23 to 75 mg GAE/100 g DW, respectively [28,35].

AOA depends mainly on the extracted compounds, which, in turn, depend on the maturity status. The results published, in this sense, are in agreement with this study. According to Haider et al. [8], the AOE was higher at khalal and thereafter but declined at the tamr stage (2.14–0.36). Allaith et al. [35] explained this decrease in AOE by the decreasing amounts of tannins, ascorbic acid and β-carotene. Awad et al. [36] explained the highest AOE during the khalal stage by the higher level of TPC. 

Likewise, AOA relies deeply on the solvent of extraction. In this context, our results are supported by the study of Hachani et al. [27], who reported that the DPPH scavenging activity of methanolic extracts was proven to be less active than acetonic extracts, with EC 50 values ranging from 0.35 to 3.70 g of dry residue/L. Similarly, a lower reducing power was obtained in 80% methanol; it varied from 2.25 to 25.68 g of dry residue/L. Additionally, Mrabet et al. [17] reported high AOA in TE and that SET enhanced it better than ST.

It can easily be seen that our cultivars exhibited potent AOA and, thus, they may have many pharmaceutical applications.

### 3.7. Correlations

The correlation coefficients (R^2^) between TPC, the sum of the individual phenolic content (IPC), TSC, HMF, the main phenolic acid (gallic acid), the main flavonoid (catechin) and the AOA as assessed by DPPH, ABTS and FRAP tests were calculated and the results are presented in Table 3. Significant positive correlations (5%) were pointed out between TPC and FRAP as well as ABTS scavenging capacity. Nevertheless, the TPC was negatively correlated with DPPH. A similar trend of correlations was obtained between the IPC and the AOA as expressed by ABTS, DPPH and FRAP, implying that phenolics are the main components responsible for the AOA of date fruits. The findings are in line with those of other authors who highlighted the potent contribution of phenolic compounds to the antioxidant capacities of date fruits [31,33,35].

Additionally, gallic acid was positively correlated with ABTS (0.80) and FRAP (0.44). These correlations indicated that gallic acid and catechin did not contribute directly to the DPPH or ABTS scavenging capacity, respectively. Previous studies mentioned negative correlations between AOA and some phenols. Moreover, the strongest correlation (0.98) was observed between the TSC and HMF, which confirms the derivation of HMF from sugar.

## 4. Conclusions

The effect of the ripening stage and the extraction solvent on the phenolic and sugar profiles as well as the AOA of Algerian secondary dates was investigated. The obtained results showed that the TSC increased from the rutab to the tamr stage, whereas AOA and TPC start decreasing gradually as the fruits ripened. Among the used solvents, the hydrothermal extracts gave the best results for all the analyzed parameters. This study also revealed the richness of the Algerian secondary date cultivars in many nutrients and bioactive compounds that possess health benefits and, thus, their potential nutraceutical effects. Relying upon the obtained results, thermal treatment of date fruits is very useful for their valorization and it is a promising technology in food industries. This “alicament” can be both used as a functional food ingredient and incorporated into other formulations to obtain more healthy and nutritious foods. 

## Figures and Tables

**Figure 1 foods-10-00503-f001:**
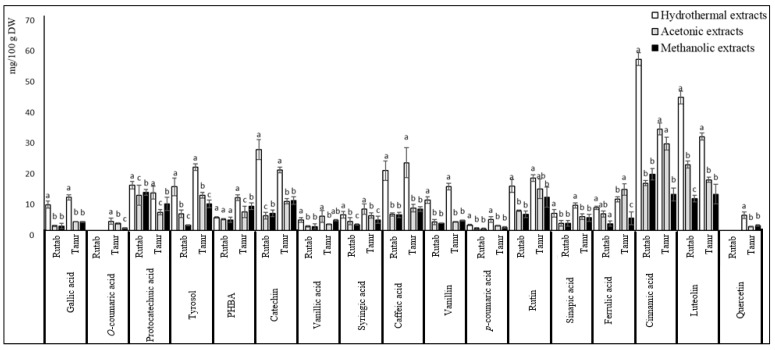
Phenolic profile in all varieties and stages for the hydrothermal extract. Values are mean ± SD. Different letters indicate that the results are statistically different (*p* < 0.05, *b* < *a*). PHBA = phosphohydroxibenzoic acid.

**Figure 2 foods-10-00503-f002:**
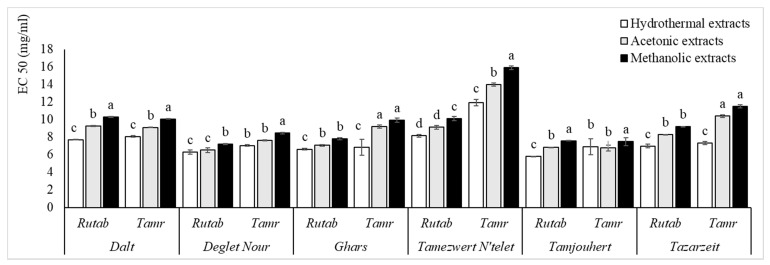
The 1,1-Diphenyl-2-picrylhydrazyl (DPPH) scavenging capacity of all cultivars and ripe stages for the three extraction systems. Values are mean ± SD. Different letters indicate that the results are statistically different (*p* < 0.05, *b* < *a*).

**Figure 3 foods-10-00503-f003:**
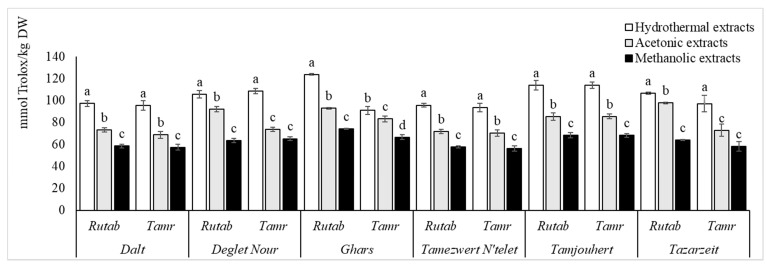
2,2′-azino-bis(3-ethylbenzothiazoline-6-sulfonic acid (ABTS) scavenging capacity of all cultivars and ripe stages for the three extraction systems. Values are mean ± SD. Different letters indicate that the results are statistically different (*p* < 0.05, *b* < *a*).

**Figure 4 foods-10-00503-f004:**
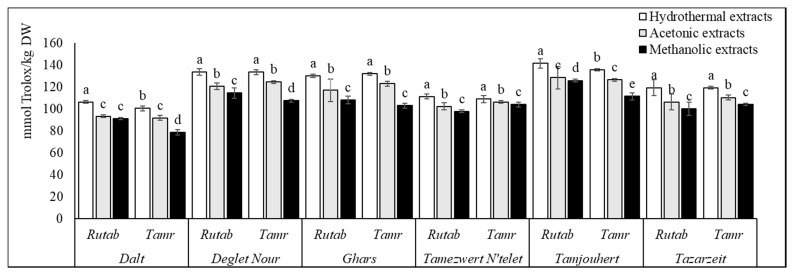
Ferric reducing power of all cultivars and ripe stages for the three extraction systems. Values are mean ± SD. Different letters indicate that the results are statistically different (*p* < 0.05, *b* < *a*).

**Table 1 foods-10-00503-t001:** Total sugar and uronic acid content in all varieties and stages for the three extraction systems.

		Extracts		
Stage	Sample	Hydrothermal	Acetonic	Methanolic
Rutab	Dalt	59.88 ± 8.11 ^c^	73.52 ± 4.70 ^b^	80.73 ± 5.37 ^a^
	Deglet Nour	55.82 ± 3.32 ^c^	76.81 ± 9.31 ^b^	82.45 ± 7.87 ^a^
	Ghars	59.33 ± 1.45 ^c^	73.94 ± 3.28 ^b^	81.58 ± 10.72 ^a^
	Tamezwert N’telet	51.42 ± 2.45 ^c^	67.78 ± 7.85 ^b^	75.91 ± 2.22 ^a^
	Tamjouhert	43.29 ± 4.43 ^b^	77.52 ± 4.70 ^a^	82.35 ± 5.96 ^a^
	Tazarzeit	26.12 ± 1.20 ^c^	78.89 ± 6.35 ^b^	88.59 ± 3.34 ^a^
Tamr	Dalt	61.24 ± 1.33 ^b^	80.15 ± 2.42 ^a^	82.69 ± 7.86 ^a^
	Deglet Nour	62.15 ± 3.42 ^b^	82.09 ± 3.44 ^a^	85.5 ± 5.54 ^a^
	Ghars	62.51 ± 3.26 ^b^	81.04 ± 5.14 ^a^	83.73 ± 1.54 ^a^
	Tamezwert N’telet	61.82 ± 4.55 ^c^	71.52 ± 3.55 ^b^	79.35 ± 9.26 ^a^
	Tamjouhert	52.32 ± 3.72 ^b^	82.19 ± 7.88 ^a^	84.72 ± 4.01 ^a^
	Tazarzeit	42.24 ± 3.82 ^b^	83.26 ± 4.33 ^a^	85.89 ± 10.2 ^a^
Uronic Acids Content (mg/100 mg DW)				
Rutab	Dalt	14.01 ± 0.89 ^a^	3.82 ± 0.33 ^b^	2.27 ± 0.55 ^b^
	Deglet Nour	9.92 ± 0.72 ^a^	5.34 ± 0.36 ^b^	4.8 ± 1.27 ^b^
	Ghars	10.99 ± 0.68 ^a^	3.73 ± 0.16 ^b^	2.87 ± 0.34 ^b^
	Tamezwert N’telet	15.57 ± 1.12 ^f^	8.6 ± 0.74 ^e^	3.48 ± 0.87 ^a^
	Tamjouhert	11.49 ± 0.92 ^a^	4.48 ± 0.17 ^c^	7.2 ± 0.58 ^b^
	Tazarzeit	6.74 ± 0.15 ^a^	5.71 ± 0.15 ^A^	5.13 ± 0.15 ^a^
Tamr	Dalt	14.35 ± 0.31 ^a^	6.12 ± 0.59 ^b^	6.63 ± 0.42 ^b^
	Deglet Nour	11.02 ± 1.26 ^a^	12.66 ± 0.17 ^a^	6.91 ± 0.97 ^b^
	Ghars	12.53 ± 1.1 ^a^	8.77 ± 0.67 ^b^	11.65 ± 0.37 ^a^
	Tamezwert N’telet	8.12 ± 0.53 ^a^	7.38 ± 0.77 ^a^	4.98 ± 0.38 ^b^
	Tamjouhert	13.87 ± 1.73 ^b^	16.33 ± 0.9 ^a^	16.28 ± 2.07 ^a^
	Tazarzeit	3.15 ± 0.56 ^b^	8.67 ± 0.6 ^a^	4.53 ± 0.37 ^b^
Total Phenolic Content (mg/100 mg DW)				
Rutab	Dalt	492.21 ± 13.04 ^a^	146 ± 0.90 ^b^	133.58 ± 8.23 ^c^
	Deglet Nour	548.46 ± 23.83 ^a^	259.24 ± 26.16 ^b^	204.44 ± 5.49 ^c^
	Ghars	526.43 ± 5.88 ^a^	213.44 ± 9.5 ^b^	188.47 ± 7.84 ^c^
	Tamezwert N’telet	424.86 ± 11.05 ^a^	191.28 ± 2.83 ^b^	160.98 ± 13.14 ^c^
	Tamjouhert	606.92 ± 63.88 ^a^	300.13 ± 1.97 ^b^	251.94 ± 3.59 ^b^
	Tazarzeit	478.85 ± 26.10 ^a^	247.84 ± 16.52 ^b^	188.74 ± 10.77 ^c^
Tamr	Dalt	492.21 ± 13.04 ^a^	146 ± 0.90 ^b^	133.58 ± 8.23 ^b^
	Deglet Nour	548.46 ± 23.83 ^a^	259.24 ± 26.16 ^b^	204.44 ± 5.49 ^b^
	Ghars	526.43 ± 5.88 ^a^	213.44 ± 9.5 ^b^	188.47 ± 7.84 ^c^
	Tamezwert N’telet	424.86 ± 11.05 ^a^	191.28 ± 2.83 ^b^	160.98 ± 13.14 ^b^
	Tamjouhert	606.92 ± 63.88 ^a^	300.13 ± 1.97 ^b^	251.94 ± 3.59 ^b^
	Tazarzeit	478.85 ± 26.10 ^a^	247.84 ± 16.52 ^b^	188.74 ± 10.77 ^c^

Values are mean ± SD. For each line, different letters indicate that the results are statistically different (*p* < 0.05, *b* < *a*). DW—dry weight.

**Table 2 foods-10-00503-t002:** Individual sugar (mg/kg DW) in all varieties and stages for the three extraction systems.

	Stage	Sample	Ram	Fuc	Ara	Xyl	Man	Gal	Glu	Total
Hydrothermal	Rutab	Dalt	2.47 ± 0.07 ^c^	nd	5.17 ± 0.32 ^b^	1.82 ± 2.36 ^c^	2.43 ± 0.21 ^c^	nd	19.03 ± 2.61 ^a^	30.93 ± 5.68
		Deglet Nour	2.52 ± 0.27 ^c^	nd	8.46 ± 2.64 ^b^	5.42 ± 2.02 ^b^	3.18 ± 1.49 ^c^	0.64 ± 0.83 ^d^	32.08 ± 0.27 ^a^	52.29 ± 9.87
		Ghars	2.66 ± 0.13 ^c^	nd	nd	4.92 ± 0.53 ^b^	nd	nd	12.70 ± 1.48 ^a^	20.28 ± 0.93
		Tamezwert N’telet	2.95 ± 0.19 ^b^	nd	5.58 ± 0.02 ^b^	3 ± 3.77 ^b^	13.94 ± 2.65 ^a^	nd	11.11 ± 1.48 ^a^	36.57 ± 8.11
		Tamjouhert	2.4 ± 0.47 ^b^	1.86 ± 0.35 ^b^	4.74 ± 0 ^b^	3.69 ± 0.03 ^b^	2.52 ± 0.02 ^b^	nd	26.88 ± 2.15 ^a^	42.09 ± 3.02
		Tazarzeit	2.34 ± 0.5 ^c^	1.16 ± 0.05 ^c^	7.1 ± 2.56 ^b^	4.6 ± 0.39 ^c^	9.5 ± 2.18 ^b^	0.72 ± 0.08 ^d^	12.35 ± 0.46 ^a^	37.77 ± 6.22
	Tamr	Dalt	2.54 ± 0.06 ^b^	nd	nd	nd	3.11 ± 0.02 ^b^	nd	61.45 ±5.40 ^a^	67.10 ± 5.47
		Deglet Nour	2.48 ± 0.09 ^b^	nd	5.36 ± 0.17 ^b^	4.03 ± 0.05 ^b^	2.89 ± 0.9 ^b^	nd	79.83 ± 7.66 ^a^	94.59 ± 8.87
		Ghars	2.32 ± 0.02 ^b^	nd	nd	nd	2.15 ± 0.12 ^b^	nd	58.54 ± 1.15 ^a^	63.01 ± 1.30
		Tamezwert N’telet	2.69 ± 0.26 ^b^	1.26 ± 0 ^b^	nd	nd	2.28 ± 0.15 ^b^	nd	53.18 ± 2.85 ^a^	59.41 ± 3.26
		Tamjouhert	2.25 ± 0 ^b^	nd	5.2 ± 0.05 ^b^	3.93 ± 0.04 ^b^	2.17 ± 0.02 ^b^	nd	115.85 ± 9.22 ^a^	129.39 ± 9.33
		Tazarzeit	2.35 ± 0.05 ^a^	nd	nd	3.66 ± 0.01 ^a^	2.31 ± 0.11 ^a^	nd	135.16 ± 31.52 ^a^	143.49 ± 31.69
Acetonic	Rutab	Dalt	4.28 ± 0.11 ^c^	nd	8.49 ± 0.87 ^b^	2.99 ± 3.87 ^c^	3.77 ± 0.71 ^c^	0.29 ± 0.37 ^d^	51.61 ± 5.55 ^a^	71.42 ± 0.38
		Deglet Nour	3.34 ± 0.57 ^c^	0.96 ± 1.25 ^d^	8.06 ± 0.49 ^b^	6.63 ± 0.79 ^b^	9.52 ± 0.05 ^b^	1.23 ± 0.46 ^c^	62.87 ± 4.99 ^a^	92.62 ± 4.02
		Ghars	4.41 ± 0.36 ^d^	2.48 ± 0.26 ^d^	8.95 ± 0.84 ^c^	7.09 ± 0.59 ^c^	29.97 ± 1.13 ^b^	nd	63.10 ± 1.53 ^a^	116.00 ± 1.00
		Tamezwert N’telet	4.12 ± 0.18 ^d^	2.75 ± 0.06 ^d^	23.74 ± 18.63 ^b^	23.57 ± 0.89 ^b^	7.25 ± 1.36 ^c^	9.48 ± 2.26 ^c^	62.89 ± 0.19 ^a^	133.80 ± 14.03
		Tamjouhert	1.43 ± 2.02 ^d^	3.61 ± 0.16 ^d^	19.07 ± 1.98 ^c^	12.24 ± 0.99 ^c^	49.79 ± 15.89 ^b^	nd	78.98 ± 4.80 ^a^	165.13 ± 12.02
		Tazarzeit	3.77 ± 0.05 ^d^	1.96 ± 0 ^d^	8.86 ± 0.33 ^c^	6.24 ± 0.15 ^c^	20.52 ± 5.16 ^b^	nd	98.50 ± 0.42 ^a^	139.86 ± 6.12
	Tamr	Dalt	3.78 ± 0.54 ^c^	nd	8.23 ± 0.81 ^b^	6.29 ± 0.2 ^b^	4.17 ± 0.65 ^c^	nd	168.46 ± 4.13 ^a^	172.22 ± 4.64
		Deglet Nour	3.59 ± 0.75 ^c^	1.68 ± 2.17 ^d^	7.78 ± 0.08 ^b^	2.99 ± 3.87 ^c^	10.59 ± 2.6 ^b^	nd	235.14 ± 3.39 ^a^	261.77 ± 4.63
		Ghars	1.59 ± 2.25 ^d^	1.98 ± 2.54 ^d^	9.93 ± 0.68 ^c^	15.03 ± 1.02 ^b^	10.82 ± 4.76 ^c^	nd	222.67 ± 16.11 ^a^	262.04 ± 8.47
		Tamezwert N’telet	4.28 ± 0.22 ^c^	1.95 ± 0.01 ^d^	7.98 ± 0.23 ^b^	5.44 ± 0.72 ^c^	4.41 ± 5.52 ^c^	nd	192.41 ± 2.10 ^a^	216.47 ± 4.58
		Tamjouhert	3.65 ± 0.3 ^d^	0.91 ± 1.18 ^e^	7.64 ± 0.28 ^c^	6.29 ± 0.48 ^c^	10.29 ± 0.67 ^b^	nd	393.99 ± 36.38 ^a^	422.78 ± 38.73
		Tazarzeit	4.33 ± 0.18 ^c^	nd	8.56 ± 0.85 ^b^	3.56 ± 4.51 ^c^	4.53 ± 0.53 ^c^	nd	677.59 ± 7.60 ^a^	698.57 ± 2.61
Methanolic	Rutab	Dalt	5.13 ± 1.51 ^d^	nd	12.5 ± 0.01 ^c^	10.33 ± 1.61 ^c^	15.3 ± 12.11 ^b^	nd	58.51 ± 1.43 ^a^	93.02 ± 16.67
		Deglet Nour	6.24 ± 0 ^c^	nd	11.6 ± 0 ^b^	nd	7.5 ± 0 ^c^	nd	57.63 ± 4.06 ^a^	76.82 ± 4.06
		Ghars	7.67 ± 0.24 ^e^	3.3 ± 3.81 ^f^	36.23 ± 14.06 ^c^	12.83 ± 3.12 ^d^	48.74 ± 37.28 ^b^	3.5 ± 4.04 ^f^	78.41 ± 10.96 ^a^	155.59 ± 73.51
		Tamezwert N’telet	4.86 ± 0.6 ^d^	nd	11.82 ± 0.06 ^b^	8.89 ± 0.3 ^c^	18.6 ± 0.01 ^b^	nd	68.56 ± 14.69 ^a^	100.10 ± 15.66
		Tamjouhert	5.2 ± 0 ^d^	4.4 ± 0 ^d^	11.88 ± 0 ^c^	10.48 ± 0 ^c^	46.21 ± 0 ^b^	nd	78.21 ± 1.63 ^a^	144.80 ± 1.63
		Tazarzeit	8.6 ± 2.19 ^d^	1.94 ± 2.48 ^e^	12.85 ± 0.38 ^c^	12.76 ± 3.99 ^c^	30.03 ± 12.88 ^b^	0.95 ± 1.21 ^e^	147.75 ± 0.63 ^a^	194.24 ± 23.77
	Tamr	Dalt	8.24 ± 3.64 ^c^	2 ± 2.83 ^d^	17.21 ± 4.41 ^b^	nd	19.1 ± 11.75 ^b^	nd	229.86 ± 6.9 ^a^	276.41 ± 29.53
		Deglet Nour	6.55 ± 0.62 ^d^	nd	13.57 ± 2.14 ^c^	10.99 ± 1.12 ^c^	33.81 ± 18.97 ^b^	nd	312.42 ± 1.86 ^a^	377.34 ± 24.70
		Ghars	8.13 ± 1.88 ^c^	3.69 ± 0.7 ^a^	11.72 ± 0.06 ^b^	9.49 ± 0.97 ^c^	14.01 ± 0.93 ^b^	nd	117.91 ± 0.46 ^a^	149.44 ± 5.00
		Tamezwert N’telet	5.42 ± 1.02 ^d^	5.44 ± 1.04 ^a^	12.14 ± 0.07 ^c^	10.04 ± 0.62 ^c^	18.6 ± 6.39 ^b^	nd	242.31 ± 5.57 ^a^	260.44 ± 14.70
		Tamjouhert	6.1 ± 0.44 ^e^	nd	24.34 ± 16.71 ^c^	18.86 ± 11.66 ^d^	69.61 ± 11.3 ^b^	nd	586.18 ± 24.72 ^a^	644.45 ± 64.83
		Tazarzeit	7.55 ± 0.74 ^d^	4.84 ± 0.6 ^e^	12.22 ± 0.33 ^c^	11.75 ± 0.76 ^c^	48.24 ± 0.08 ^b^	nd	489.48 ± 28.06 ^a^	514.37 ± 30.57

Values are mean ± SD. For each line, different letters indicate that the results are statistically different (*p* < 0.05, *b* < *a*). nd = not determined. Ram: Rhamnose; Fuc: Fucose; Ara: Arabinose; Xyl: Xylose; Mal: Maltose; Gal: Galactose; Glu: Glucose.

**Table 3 foods-10-00503-t003:** Correlation coefficients between phenolic compounds and antioxidant activity.

	ABTS	DPPH	FRAP	TPC	IPC	Gallic Acid	Catechin	TSC	HMF
ABTS		−0.47	0.70	0.68	0.76	0.80	−0.33	−0.24	0.29
DPPH	−0.47		−0.52	−0.43	−0.37	−0.24	0.04	0.35	−0.03
FRAP	0.70	−0.52		0.72	0.56	0.44	−0.21	−0.17	0.05
TPC	0.68	−0.43	0.72		0.90	0.57	0.05	−0.16	0.12
IPC	0.76	−0.37	0.56	0.90		0.90	−0.36	−0.37	0.31
Gallic acid	0.80	−0.24	0.44	0.57	0.90		0.24	−0.38	0.40
Catechin	−0.33	0.04	−0.21	0.05	−0.36	−0.24		0.02	−0.14
TSC	−0.24	0.35	−0.17	−0.16	−0.37	−0.38	0.02		0.98
HMF	0.29	−0.03	0.05	0.12	0.31	0.40	−0.14	0.98	

TCP = Total Phenolic Content; ICP = Individual Phenolic Content; TSC = Total Sugar Content; HMF = Hydroximethylfurfural.

## Data Availability

The authors confirm that the data supporting the findings of this study are available within the article.

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
