# Peer review of "Antioxidant Capacity and Phenolic and Sugar Profiles of Date Fruits Extracts from Six Different Algerian Cultivars as Influenced by Ripening Stages and Extraction Systems"

_foods, 2021, doi:10.3390/foods10030503_

Round 1

Reviewer 1 Report

1.  Line 29;  must re-write/indicated the stages of dale mantuation/rippen as authors indicated in Ref 1.  The authors in Ref 1 clearly explained four stages of maturation date, Not three as you indicated in this manuscript.  This is importance for later in this manuscript.

The Authors in this manuscript must change all the stage of date that used in this manuscript.   rutab = full-ripe  and tamar = semidry or dry.  These information must be change entlie in this manuscript.   It was not  unripe and ripe as authors indicated in whole manuscript.

Line 69;  .....homogenized with 20mL of the solvent.....

                   must indicated what type of solvent that was used.

2.  Line 70-74;  must re-write such as how many volume of each experiment was used such as  volume mL of MeOH:H2O, volume mL of Acetone:H2O and volume of H2O.   Authors must  provide detail of each method that used to extract date samples.   Author must use consistency word for each extraction process such water extract that actual it is hydrothermal extraction.

Thus authors must use consistency abbreviation  MeOH:H20 extraction = ME, Acetone:H2O extraction = AE and Hydrothermal extraction = HE and conuse for entire  of manuscript

Line 73; provide the reason why kept extraction at 4C and for how long before analysis.

3.  Make sure that Authors must provided abbreviation before used it in manuscript such as  3.1 Total sugar content (TSC), 3.2 Uronic acids content (UAC)

4.  Provide name of each date instead of S1, S2, S3.........

5.  All tables must improve by using universal number such as 59.88 instead of 59,88

5.  Consideration to present the results in new way in Table 4-6 such as combine raw data of caultiva 1-6 of each stage for each type of phenolic compound then present in bar graph.  This will help reader to much more understand the effect of extraction protocol.

Author Response

Response to Reviewer 1´s comments:

  • Line 29;must re-write/indicated the stages of date maturation/rippen as authors indicated in Ref 1. The authors in Ref 1 clearly explained four stages of maturation date, Not three as you indicated in this manuscript. This is important for later in this manuscript.

Response: The sentence has been rewritten and the four stages described in Ref 1 have been cited in the revised manuscript (Line 39 – 40).

  • The Authors in this manuscript must change all the stage of date that is used in this manuscript. rutab = full-ripe and tamar = semidry or dry. These information must be change entlie in this manuscript. It was not unripe and ripe as authors indicated in whole manuscript.

Response: The previously cited stages have been changed and the denominations “tamr” and “rutab” are used along the revised paper (text, tables and figures).

  • Line 69; .....homogenized with 20mL of the solvent…, must indicated what type of solvent that was used.

Response: The revised version includes the type of the used solvents (Line 84).

  1. Line 70-74; must re-write such as how many volume of each experiment was used such as volume mL of MeOH:H2O, volume mL of Acetone:H2O and volume of H2O. Authors must provide detail of each method that used to extract date samples. Author must use consistency word for each extraction process such water extract that actual it is hydrothermal extraction. Thus authors must use consistency abbreviation MeOH:H20 extraction = ME, Acetone:H2O extraction = AE and Hydrothermal extraction = HE and confuse for entire of manuscript

Response: The details of each method of extraction have been mentioned in the revised paper (Lines 80 – 89).

Line 73; provide the reason why kept extraction at 4C and for how long before analysis.

Response: Actually, the extracts were kept at 4°C to avoid microbial contamination; noticing that where the period between storage and analysis did not exceed one week.

  1. Make sure that Authors must provided abbreviation before used it in manuscript such as 3.1 Total sugar content (TSC), 3.2 Uronic acids content (UAC)

Response: The revised paper includes abbreviations whose initials have been cited previously (Lines 91, 97, 96, 108, 144 and 304).

  1. Provide name of each date instead of S1, S2, S3.........

Response: The abbreviations S1,S2 and S3 have been replaced by the name of each cultivar (Tables 1 - 2).

  1. All tables must improve by using universal number such as 59.88 instead of 59,88

Response: Tables have been revised so that all the numbers are universal (Tables 1, 2 and 3).

  1. Consideration to present the results in new way in Table 4-6 such as combine raw data of cultivars 1-6 of each stage for each type of phenolic compound then present in bar graph. This will help reader to much more understand the effect of extraction protocol.

Response: As suggested, tables 3-6 have been converted to bar graphs for better understanding the effect of extraction protocol. (Figures 1, 2, 3 and 4).

Reviewer 2 Report

Submitted manuscript is interesting and well written. The introduction seems quite short considering the length of the article, however it is sufficient and requires only minor inclusion (details in comments). The methods section is quite detailed, however some important information should also be included. The strongest part of the manuscript is results and discussion – including many interesting results and proper discussion with reference to literature data. Conclusions are properly drawn. The style of the manuscript requires some minor changes and references are not properly formatted, but it does not influence the scientific value of the manuscript.

Detailed comments:

Line 29 please describe briefly differences between those stages

Line 30-32 please rephrase, divide or extend the sentence, in current form it is hard to read

Line 57-58 please state purity

Line 66 please add some more basic details about extraction procedure i.e.  technique and time

Line 71-72 please use term sterilized

Line 160 up then is not proper in this case

Table 2 Not Ripe is written with capital letters at the beginning whereas ripe is not

Line 354-370 Please place table with correlation coefficient and p value for all mentioned correlations.

Author Response

Response to the Reviewer 2´s comments:

  • Line 29 please describe briefly differences between those stages

Response: The revised document includes brief description of each stage (Lines 39 – 40).

  • Line 30-32 please rephrase, divide or extend the sentence, in current form it is hard to read

Response: The sentence has been rephrased for better understanding (Lines 27– 30).

  • Line 57-58 please state purity

Response: A list of all the chemicals as well as their relative purity has been added under your request and that of another reviewer (Lines 63 – 72).

  • Line 66 please add some more basic details about extraction procedure i.e. technique and time

Response: The details of the extraction procedure (technique, time, solvents, purity, ratio…) are given in the revised paper (Lines 80 – 89).

  • Line 71-72 please use term sterilized

Response: The word “autoclaved” had been replaced by the word “sterilized” (Line 86).

Line 160 up then is not proper in this case

Response: The word “up then” had been replaced by the word “above” (Line 179).

Table 2 Not Ripe is written with capital letters at the beginning whereas ripe is not

Response: The format of all the tables had been revised (Table 1, 2 and 3).

Line 354-370 Please place table with correlation coefficient and p value for all mentioned correlations.

Response: The correlation table and the p value have been added (Line 380, table 3).

Reviewer 3 Report

Dear editor, the ms provides a phenolics and sugars characterization in date fruits by HPLC and UV Vis methods. 

The article presents interesting research with good aim to be reached with the tool utilized but it has several flaws.

  • Lines 19 and 20 are in contradiction, at first was declared that the sugars don't depend on the ripe stage, and then that galactose was present only in unripe samples. This evidence a possible influence on galactose formation during ripening. Moreover, the same contradiction is reported in the text. 
  • The introduction has only 5 references, too few to explain the paper background.
  • Section 2.1. A specific list of all the chemicals used has to be added.
  • Section 2.2. Why did the authors use tamr and rutab words and the ripe and unripe? I think this has to be explained. I suggest using the English notation and only to mention the local one. 
  • Section 2.2. How many samples have been involved in the study? 
  • Section 2.3. Why did the author use the extraction phases at that concentration and reported others in introduction and results discussions? MeOH:H20 65:30 vs 80:20... please justify the choice and provide a reference.
  • Lines 108, what's the mean of "tentatively identified"?
  • Lines 153- 157, in these sentences the authors stated that sugar content increases during ripening in contradiction to abstract (as above-mentioned).
  • Lines 174 - 177, how the uronic acids are related to lignin and pectin? Please explain or add references.
  • Line 201, widely studied and just one reference?
  • Lines 203-219, this paragraph seems like a review and no a comparison.
  • Lines 227-228, how the authors can compare results in FW and DW? This makes no sense. 
  • Lines 262, what's the means of "more ore less previously quantified"? This is no a scientific soundness.
  • Table 3, Table 4, and Table 5. "Traces = very low amount"? What does it mean? This is no a scientific notation. You can use not identified, not detected, under LOQ, under LOD, but a very low amount makes no sense.
  • Section 3.6 Are the Correlation statistically significant? 
  • I suggest the use of multivariate data analysis, even if explorative one as Principal Component Analysis, to better explore the data and data correlation.

Author Response

Response to the Reviewer 3´s comments:

  • Lines 19 and 20 are in contradiction, at first was declared that the sugars don't depend on the ripe stage, and then that galactose was present only in unripe samples. This evidence a possible influence on galactose formation during ripening. Moreover, the same contradiction is reported in the text.

Response: Actually, it was not a contradiction it was a typing error. The sentence has been revised so that to rectify the error and hence avoid contradiction (Lines 18 – 19).

  • The introduction has only 5 references, too few to explain the paper background.

Response: Reviewing literatures have been updated and the findings of some new articles were added in the introduction to enrich the paper background (Lines 26 – 62).

  • Section 2.1. A specific list of all the chemicals used has to be added.

Response: The list of chemicals has been added (Lines 63 - 72).

  • Section 2.2. Why did the authors use tamr and rutab words and the ripe and unripe? I think this has to be explained. I suggest using the English notation and only to mention the local one.

Response: The only words used in revised version to refer to the two ripening stages are “tamr” and “rutab” the other words have been deleted to avoid any misunderstanding.

  • Section 2.2. How many samples have been involved in the study?

Response: From different lots and date palm bunches, five (5) representative samples of about 1 kg each one had been collected from each cultivar and each stage.

  • Section 2.3. Why did the author use the extraction phases at that concentration and reported others in introduction and results discussions? MeOH:H20 65:30 vs 80:20... please justify the choice and provide a reference.

Response: Actually, the choice of solvents and their concentrations was done based on several studies. Nevertheless, their authors used different techniques, solvents, concentrations and polarities and evidently obtained different results. Benkerrou et al., (2018) revealed that extraction with acetone concentration of 66.71% (v/v) is ideal to optimize the extraction of date fruit antioxidants. Mrabet et al., (2017). reported an increase of 15 % in hydrothermal extracts. Zoreky & Al–Tahar (2015) indicated that the 80% methanolic extract was the best extractor. Hence, we have chosen aqueous extraction and organic one using polar solvents (unpublished preliminary studies revealed that unipolar one are not).

Regarding the introduction, result and discussion we relied on the work of Zoreky & Al–Tahar (2015)) because, like us, they emphasized the effect of the solvent; even though they used different concentration (MeOH:H20 65:30 vs 80:20)and  without neglecting the other results.

  • Lines 108, what's the mean of "tentatively identified"?

Response: The identification of the phenolic acids and flavonoids was done not only by comparing samples’ retention time and UV spectra to standards but also by running the samples after the addition of pure standards. This word has been deleted (Line 123).

  • Lines 153- 157, in these sentences the authors stated that sugar content increases during ripening in contradiction to abstract (as above-mentioned).

Response: We reinforce that sugar content increases during ripening and the sentence was rephrased to avoid contradiction as mention below in the answer of the first comment.

  • Lines 174 - 177, how the uronic acids are related to lignin and pectin? Please explain or add references.

Response: It is admitted that galacturonic acid (galactose + uronic acid) is the main component of pectin. Regarding lignin, it is not directly related to uronic acid. Actually, to establish a relationship between date fruit quality and its uronic acid, we shall pass through the contents of lignin and pectine. As mentioned in the paper, the date fruit’s good quality results from high amounts of pectin and low amounts of lignin.

  • Line 201, widely studied and just one reference?

Response: We have cited more than one reference in the revised paper to discuss this section. References 1, 13, 14, 16, 17 and 19 carried all about date fruits sugar profile (Lines 219 – 232).

  • Lines 203-219, this paragraph seems like a review and no a comparison.

Response: The paragraph was rephrased in order to get a comparative discussion (Lines 219 – 232).

  • Lines 227-228, how the authors can compare results in FW and DW? This makes no sense.

Response: In the revised manuscript, we have compared our results to newer ones, expressed both in DW (Lines 237 – 242).  

  • Lines 262, what's the means of "more ore less previously quantified"? This is no a scientific soundness.

Response: We used the expression “more or less previously quantified” to mention that some authors, for example Mansouri et al., 2005, did not quantify the identified compounds. However, it has been deleted to overcome any confusion (Line 273).

  • Table 3, Table 4, and Table 5. "Traces = very low amount"? What does it mean? This is no a scientific notation. You can use not identified, not detected, under LOQ, under LOD, but a very low amount makes no sense.

Response: The mentioned tables have been converted into bar graphs and word “traces” does not disturb the readers any more.

  • Section 3.6 Are the Correlation statistically significant?

Response: The correlation coefficients are statistically studied at 5 %, the correlation table and the p value are added as table 3 (Line 380).

  • I suggest the use of multivariate data analysis, even if explorative one as Principal Component Analysis, to better explore the data and data correlation.

Response: Actually, the ANOVA test was performed (5%) to compare between the different cultivars, stages and solvents. The T-Tukey test allowed us classifying samples into groups. On the other hand, the Principal Component Analysis (PCA) served to visualize the correlation between the bioactive compounds and antioxidant capacity of our samples (Line 150 – 157).

Round 2

Reviewer 1 Report

line 87;  hydrothermal extracts (TE)  change to hydrothermal extracts (HE).

Line 88;  stored at 4C for further analysis  change to stored at 4C and processed analysis within 1 week.

Table 2.   Move Methanol  to next page so the pattern will be the same as Hydrothermal and Acetonic

Table 2.   All name of sample must use the same pattern such as if authors prefer start with lower case, so use all name start with lower case;  dalt, Deglet nour.   If authors prefer start with upper case, so use all name start with upper case (Dalt, Deglet nour...) in every tables and Figures

All Figures:  they are looked more confused.

                   Pleas change to white bar for HE, gray bar for AE and black bar 

                            for ME   Then Move all these to Lt or Rt top of conner of the graph

Authors must read and check all manuscript again: the authors still leave such as  0,6-4,5% in line 278; 0,4+/- 0,0 in line 302 instead of 0.6-4.5%

Author Response

Response to de Reviewer 1´s comments :

  • line 87; hydrothermal extracts (TE) change to hydrothermal extracts (HE).

Response: The abbreviation TE has been changed to HE (Lines 87, 187, 235, 262, 343, 344, 345)

  • Line 88;stored at 4C for further analysis change to stored at 4C and processed analysis within 1 week.

Response: The expression “stored at 4°C for further analysis” has been  changed to “stored at 4C and processed analysis within 1 week” (Line 88).

  • Table 2. Move Methanol to next page so the pattern will be the same as Hydrothermal and Acetonic

Response: The results of methanolic extracts have been moved to next page (Table 2; Page 8)

  • Table 2. All name of sample must use the same pattern such as if authors prefer start with lower case, so use all name start with lower case; dalt, Deglet nour. If authors prefer start with upper case, so use all name start with upper case (Dalt, Deglet nour...) in every tables and Figures

Response: The format of the all the tables and figures has been reviewed so that the samples’ names are written similarly.

  • All Figures: they are looked more confused. Pleas change to white bar for HE, gray bar for AE and black bar for ME. Then Move all these to Lt or Rt top of conner of the graph.

Response: The bars’ color corresponding to HE, AE and ME has been changed to white, gray for and black respectively and all of them have been moved to the right conner of the graph.

  • Authors must read and check all manuscript again: the authors still leave such as 0,6-4,5% in line 278; 0,4+/- 0,0 in line 302 instead of 0.6-4.5%

Response: The manuscript has been check so that all the used numbers are universal (Lines 213, 279, 304).

Thanks to the reviewer's comments, it has been possible to improve the quality of the manuscript.

Reviewer 3 Report

The authors improved the ms with all my queries

Author Response

Thanks to the reviewer's comments, it has been possible to improve the quality of the manuscript.